# The Pro-Apoptotic Effect of Silica Nanoparticles Depends on Their Size and Dose, as Well as the Type of Glioblastoma Cells

**DOI:** 10.3390/ijms22073564

**Published:** 2021-03-30

**Authors:** Rafał Krętowski, Magdalena Kusaczuk, Monika Naumowicz, Marzanna Cechowska-Pasko

**Affiliations:** 1Department of Pharmaceutical Biochemistry, Medical University of Bialystok, 15-089 Bialystok, Poland; r.kretowski@umb.edu.pl (R.K.); mkusaczuk@wp.pl (M.K.); 2Institute of Chemistry, University in Bialystok, 15-328 Bialystok, Poland; monikan@uwb.edu.pl

**Keywords:** apoptosis, glioblastoma, nanotoxicity, silica nanoparticles

## Abstract

Despite intensive investigations, nanoparticle-induced cellular damage is an important problem that has not been fully elucidated yet. Here, we report that silica nanoparticles (SiNPs) demonstrated anticancer influence on glioblastoma cells by the induction of apoptosis or necrosis. These effects are highly cell type-specific, as well as dependent on the size and dose of applied nanoparticles. Exposure of LN-18 and LBC3 cells to different sizes of SiNPs—7 nm, 5–15 nm, or 10–20 nm—at dosages, ranging from 12.5 to 1000 µg/mL, for 24 and 48 h reduced the viability of these cells. Treatment of LN-18 and LBC3 cells with 7 nm or 10–20 nm SiNPs at doses ≥50 µg/mL caused a strong induction of apoptosis, which is connected with an increase of intracellular reactive oxygen species (ROS) production. The 5–15 nm SiNPs exhibited distinct behavior comparing to silica nanoparticles of other studied sizes. In contrast to LBC3, in LN-18 cells exposed to 5–15 nm SiNPs we did not observe any effect on apoptosis. These nanoparticles exerted only strong necrosis, which was connected with a reduction in ROS generation. This suggests that SiNPs can trigger different cellular/molecular effects, depending on the exposure conditions, the size and dose of nanoparticles, and cell type of glioblastoma.

## 1. Introduction

Glioblastoma represents a large category of brain tumors, and constitutes about 80% of malignant brain tumors. Due to its multiple forms, it is also termed glioblastoma multiforme (GBM). It arises from astrocytes and is the grade IV type of tumor, which is infiltrative and undifferentiated from other normal cells [1,2]. The average patient survival after diagnosis is roughly one year [3,4]. These tumors generate a lot of clinical problems. They are associated with poor prognosis, because of their site within the brain, which greatly entangles surgical ablation, aggressive character, infiltrative growth, progression to malignancy, and resistance to chemotherapy or radiotherapy [5]. Accordingly, searching for the novel methods of glioblastoma therapy with reduced undesirable effects remains necessary.

One of the latest possibilities in experimental therapeutics is the application of nanotechnology for cancer treatment [1]. Nanoparticles (NPs), which are able to transport drugs by the blood–brain barrier, are one of the most promising approaches [1,2]. Thanks to variability of physicochemical parameters, such as shape, size, structure, and elemental constituents, nanoparticles can activate multiple mode of actions in cancer cells [6]. The crucial mechanisms connected with silica nanotoxicity include aberrant aggregation of nucleoplasmic proteins, DNA damage, and formation of reactive oxygen species (ROS) [7,8,9,10]. These disorders in cellular homeostasis lead in consequence to apoptosis of damaged cells. The process of apoptotic cell death of transformed cells is a major regulator of cancer development [11]. It has already been shown that silica nanoparticles (SiNPs) may trigger apoptosis via different apoptotic pathways [12,13]. It has been demonstrated that in SiNPs-treated A549 cells, receptor-mediated apoptosis was activated [12]. Other reports, however, have indicated the role of the intrinsic pathway of apoptosis initiated in response to SiNPs exposure [14,15]. They have proven that silica nanoparticles evoked ROS formation and oxidative stress, which then cause apoptosis via the mitochondrial pathway [15], and have noticed the dose-dependent upregulation of *caspase 9* and *caspase 3* genes in A549 and A431cells [15]. In line with this, Ahmad et al. have shown the upregulation of proapoptotic *BAX* and *caspase 3*, with simultaneous downregulation of the antiapoptotic *BCL 2* gene in HepG2 cells [14]. Although apoptosis seems to be the most important process activated in response to SiNPs treatment, apoptotic cell death is not the only mechanism of cell elimination after exposure to SiNPs. Thus, SiNPs-evoked necrosis has also been documented in several studies [16,17,18]. Furthermore, it has been recently demonstrated that HUVEC cells exposed to 304 nm or 310 nm SiNPs showed increased necrosis, whereas macrophages treated with the same silica nanoparticles developed mainly apoptosis [16]. Additionally, the studies of Corbalan et al. have shown that SiNPs filtered through the membrane of endothelial cells, and quickly boosted to liberate the cytoprotective NO, and in greater degree, increased the production of cytotoxic peroxynitrite anion (ONOO-) [17]. This in consequence led to induction of nitroxidative/oxidative stress, inflammation, and necrosis [17].

At present, the exact mechanism by which SiNPs induce cytotoxicity, especially related to particle size, is still poorly understood. In order to extend this knowledge, we decided to investigate the cytotoxicity of SiNPs on two different glioblastoma cell lines (LN-18 and LBC3). Because cytotoxic alterations may be associated with the physical or chemical properties of nanoparticles, we chose distinct sizes of SiNPs (7 nm, 5–15 nm, and 10–20 nm) and examined their cytotoxic and pro-apoptotic effects, as well as production of intracellular ROS in LN-18 and LBC3 cells.

## 2. Results

### 2.1. The Effect of SiNPs on Cell Viability

The anti-proliferative effect of three diverse sizes—7 nm, 5–15 nm, or 10–20 nm—of SiNPs on LBC3 (Figure 1A–C) and LN-18 cells (Figure 2A–C) was estimated using the MTT assay. The cells were incubated with various concentrations of SiNPs (12.5 to 1000.0 µg/mL) for a period of 24 and 48 h. Our results indicate that all used silica nanoparticles to evoke a time-dependent and dose-dependent decrease of cells viability in both LBC3 and LN-18 cell lines (Figure 1A–C and Figure 2A–C).

The decrease in cell viability is dependent on sizes of silica nanoparticles, especially in the case of the LBC3 cell line. The reduction in viability of LBC3 and LN-18 cells was observed as early as after 24 h of incubation in all sizes of applied silica nanoparticles. In cells treated with higher concentrations of SiNPs, the effect on cell viability was markedly more pronounced in the case of LBC3 cells (Figure 1A–C), whereas the LN-18 cells displayed much greater viability (Figure 2A–C). We have found a size-dependent reduction in viability of LBC3 cells occurring in the following order: 5–15 > 10–20 > 7 nm (Figure 1A–C), whereas the cytotoxicity of SiNPs in LN-18 cells was not affected by their size (Figure 2A–C), in our experimental conditions.

After 48 h, we observed a strong decrease in the amount of viable SiNPs-treated LBC3 (Figure 1) and LN-18 cells (Figure 2), compared to untreated controls. Interestingly, treatment with medium-sized (5–15 nm) SiNPs reduced the viability of LBC3 cells by 80% after 24 h and by 90% after 48 h of incubation (Figure 1B).

### 2.2. The Effect of SiNPs on Apoptosis and Necrosis

The percentage of apoptotic and necrotic LBC3 and LN-18 cells in cultures treated with various concentrations of silica nanoparticles, ranging from 25 to 600 µg/mL, at three different sizes (7 nm, 515 nm or 10–20 nm), and incubated for 24 h and 48 h is presented in Figure 3, Figure 4, and Figure 5, respectively.

The treatment of LBC3 cells with 7 nm and 10–20 nm SiNPs at doses ≥ 50 µg/mL caused a significant induction of apoptosis, up to 50%, compared to the control cells. We observed a time- and dose-dependent increase in the apoptosis of this cells (Figure 3A–C and Figure 5A–C, respectively). This effect was diminished after 48 h of nanoparticle exposure (Figure 3B,C and Figure 5B,C). Interestingly, the treatment of LBC3 cells with 5–15 nm silica nanoparticles at doses ≥ 50 µg/mL was found to significantly induce a greater percentage of apoptotic cells (up to 70%), compared to the 7 nm and 10–20 nm SiNPs. We observed a time- and dose-dependent increase in the apoptosis of these cells (Figure 4A–C). This effect was more pronounced after 48 h of nanoparticle exposure (Figure 4B,C).

The percentage of necrotic LBC3 cells cultured with different concentrations (25 to 600 µg/mL) and different sizes of silica nanoparticles, for 24 and 48 h, is presented in Figure 3A,B,D; Figure 4A,B,D; and Figure 5A,B,D. Interestingly, LBC3 cells showed size-dependent changes in necrosis evoked by SiNPs in doses exceeding 50 µg/mL. Only trace amounts of necrosis in LBC3 cells treated with 7 nm SiNPs (Figure 3A,B,D) and 10–20 nm SiNPs (Figure 5A,B,D) were observed, while 5–15 nm nanoparticles caused predominantly necrotic cell death (Figure 4A,B,D).

Similar to the LBC3 cells, the treatment of LN-18 cells with 7 nm and 10–20 nm SiNPs at doses ≥50 µg/mL caused a significant induction of apoptosis, up to 25% and 60%, respectively, compared to the control cells. We observed a time- and dose-dependent increase in apoptosis of LN-18 cells exposed to 7 nm SiNPs (Figure 3E–G) and 10–20 nm SiNPs (Figure 5E–G). This effect was diminished after 48 h of nanoparticle exposure (Figure 3F,G and Figure 5F,G). In contrast to the LBC3 cells, we did not record any influence of 5–15 nm silica nanoparticles on the apoptosis of LN-18 cells. The apoptosis level in LN-18 cells did not change significantly regardless of the incubation time or doses of the silica nanoparticles.

Interestingly, the 5–15 nm silica nanoparticles exerted a strong effect on necrosis, up to 80% in LN-18 cells, in time- and dose-dependent manner (Figure 4E,F,H). The treatment of LN-18 cells with 7 nm (Figure 3E,F,H) and 10–20 nm (Figure 5E,F,G) SiNPs at the highest doses (≥100 µg/mL) caused a significant induction of necrosis, compared to the control cells. We noticed that the main mechanism initiated after treatment LN-18 cells with 5–15 nm SiNPs is necrosis, which is known as an uncontrolled process.

### 2.3. The Effect of SiNPs on Oxidative Stress

In order to estimate the relevance of oxidative stress in cytotoxicity and apoptosis induction, the formation of intracellular ROS was measured. Figure 6 and Figure 7 show the fluorescence of 2′7′-dichlorodihydrofluorescein in LBC3 and LN-18 cells treated with 50 and 100 μg/mL of 7 nm, 5–15 nm, and 10–20 nm silica nanoparticles, for 24 h and 48 h. The fluorescence of 2′7′-dichlorodihydrofluorescein was escalated with an increase in the cellular ROS production.

This research indicates that significant differences in intracellular ROS levels are size-, dose-, and cell type-dependent. Incubation of LBC3 cells with three different sizes of SiNPs resulted in enhanced intracellular ROS production (Figure 6A–C). The highest number of generated ROS was demonstrated after 48 h of 5–15 nm SiNPs exposure (Figure 6B).

In LN-18 cells, the treatment with 7 nm and 10–20 nm SiNPs resulted in increased intracellular ROS production (Figure 7A,C). Interestingly, LN-18 cells treated with 5–15 nm SiNPs at doses of 50 and 100 μg/mL showed a reduction in ROS generation (Figure 7B), which might be connected with the reduced apoptosis of these cells.

## 3. Discussion

GBM is one of the most progressive and deathly diseases. There is no efficient therapy for GBM, thus far it still remains an incurable malignancy. The most essential problem taken into account in GBM therapy is the occurrence of the blood–brain barrier and blood–brain tumor barrier, which inhibit efficient drug delivery [1,19].

Great progress in nanotechnology offers new remedies for the development of new methods in cancer diagnosis and treatment. Nanoparticles are one of the most promising proposals that are capable of breaking the blood–brain barrier [19,20]. Interestingly, initial reports suggest that SiNPs may be able to cross the blood–brain barrier, making them an interesting agent for use in glioblastoma research [21]. Compared to other nanomaterials, SiNPs, with their large specific surface area and high porosity, are promising candidates because of several advantages, such as excellent biocompatibility; highly porous structure; ability to self-assemble; tumor-specific targeting; small size; being fast, simple, and low-cost to produce; biodegradability; and facility in application [1,3,22]. Moreover, SiNPs possess residual silanol groups (Si–OH) at the surface that can be functionalized by different organic groups [23]. Additionally, it has been proven that SiNPs may penetrate cell membranes, lodge in the mitochondria, and lead to damage of cancer cells. After being endocytosed, SiNPs interact with different organelles, including endosomes, lysosomes, the ER, and the Golgi apparatus, and transport via diverse routes, such as the endolysosomal, ER/Golgi, and cytoplasmic routes, resulting in totally different destinies. More mechanism studies are needed to shed new insights on the details of SiNPs clinical usage, in order to improve glioma therapy and prognosis [23].

Because the size-dependent toxicity of SiNPs is still disputable, monodispersed SiNPs of controlled sizes were used in our study. Thus, in our study, we determined SiNPs-induced cytotoxicity. The decrease of cell viability was dependent on sizes of silica nanoparticles, and was especially pronounced in the LBC3 cell line. The reduction of LBC3 and LN-18 cell viability was noted in all sizes of SiNPs used. In cells handled with higher concentrations of silica nanoparticles, the influence on cell viability was clearly more pronounced in the LBC3 cell line, whereas the LN-18 cells displayed much greater viability. We have found size-dependent reduction in the viability of LBC3 cells to occur in the following order: 5–15 nm > 10–20 nm > 7 nm. However, the cytotoxicity of SiNPs in LN-18 cells was not impacted by their size in our experimental conditions.

A great deal of research concentrating on the toxic effects of silica nanoparticles has identified the nanoparticle surface area as one of the most important factors contributing to their toxicity [24,25]. Since the surface area of the nanoparticles is strictly dependent on their size, this parameter might be considered as a critical factor affecting cellular nanotoxicity [24,25]. Previous studies have shown contradictory effects, referring to the connection between SiNPs size and their cytotoxicity. The investigation of SiNPs with diameters of 30 nm, 48 nm, 118 nm, and 535 nm has revealed reduced viability of mouse keratinocytes with the decreasing size of nanoparticles [26]. The opposite results were reported by Lu et al. [27], who used SiNPs with diameters of 7 nm, 20 nm, and 50 nm at doses ranging from 20 to 640 µg/mL. They have found reduced viability in human hepatocytes in the following order 20 > 7 > 50 nm [27].

Nanoparticles usually exert their anticancer effects by triggering apoptosis and/or necrosis [12,28]. Thus, we determined SiNPs-induced apoptosis and necrosis. The results of Annexin V/PI staining demonstrated that apoptosis was induced by SiNPs in both LBC3 and LN-18 cells, with the exception of LN-18 cells treated with 5–15 nm SiNPs, where only necrosis was exerted.

The results from our study showed that although SiNPs could induce apoptosis or necrosis in both glioblastoma cell lines, the percentage of apoptotic or necrotic cells was different and cell type-dependent. The treatment of LBC3 cells with 7 nm and 10–20 nm SiNPs caused a significant, time- and dose-dependent induction in apoptosis (up to 50%). Interestingly, we found that exposure of LBC3 cells to 5–15 nm SiNPs induces a much higher increase in apoptosis, up to 70%, compared to the apoptosis exerted by 7 nm and 10–20 nm SiNPs. Similar to the LBC3 cells, the treatment of LN-18 cells with 7 nm and 10–20 nm SiNPs caused a significant increase in apoptosis (up to 25% and 60%, respectively), compared with the control.

As opposed to the LBC3 cells, we did not observe any influence on apoptosis in LN-18 cells treated with 5–15 nm silica nanoparticles. The amount of apoptotic LN-18 cells did not change significantly, regardless of the incubation time and doses of the silica nanoparticles. Interestingly, the SiNPs of 5–15 nm in LN-18 cells exerted extensive necrosis, up to 80%. The current study demonstrated that the pro-apoptotic effect of SiNPs depends on the type of glioblastoma cells. The difference of SiNPs action on LBC3 and LN-18 cells may be the results of the distinct *P53* status and other mutations occurring in these cells. The LN18 cells exhibit mutated *P53* (*TP53*) *TGT(Cys)* → *TCT(Ser)* (mutation at codon 238), while the LBC3 cell line exhibits a mutation of *P53* in the DNA-binding domain. Additionally, the LN18 cells exhibit homozygous deletions in the *P16* and *P14ARF* tumour-suppressor genes. They have a wild-type *PTEN* gene. Meanwhile, the LBC3 cell line was first transfected with *α9* (*LBC3α9+*) and *P75NTR* (*LBC3P75+*) separately, and then double transfected with both transmembrane receptors (*LBC3α9+/P75+*). The LBC3 cells express *PTEN*, which is constitutively phosphorylated. These findings are in agreement with the studies of Pan et al., who also demonstrated that 1.4 nm nanoparticles evoke necrosis, while 1.2 nm nanoparticles reduce increased levels of apoptosis in several human cell lines when exposed to gold nanoparticles [29].

Recently, a plethora of research concerning the relationship between apoptosis and necrosis has been published. Apoptosis and programed necrosis called “necroptosis” have potential connections as two forms of programmed cell death (PCD). Morphologically, necroptosis shows the features of necrosis, although it is initiated by the necrosome formation [28,30]. Some researchers have reported that necroptosis occurs only when apoptosis is inhibited [28,31,32]. However, it has also been mentioned that apoptosis and necroptosis may be induced simultaneously, not only by the blockade of apoptosis, but also by chemotherapeutic agents [28,31,32]. The mechanisms underlying the induction of both phenomena apoptosis and necroptosis are still unclear. Many studies have shown that autophagy might play a key role in this situation [28,30]. Given the significant role of apoptosis and necrosis/necroptosis in the inhibition of tumor cell growth, we suggest that the production of reactive oxygen species might act as a tumor suppressor mechanism, leading to oxidative stress and the induction of apoptosis or necrosis. Significant differences were found in cellular ROS levels, depending on cell type and size or dose of SiNPs. Incubation of LBC3 cells with three different sizes of SiNPs resulted in the overproduction of intracellular ROS. The highest amount of generated ROS was observed after 48 h of 5–15 nm SiNPs exposure. Similar to LBC3 cells, the treatment of LN-18 cells with 7 nm and 10–20 nm SiNPs resulted in a rise of intracellular ROS production. In contrast, LN-18 cells treated with 5–15 nm SiNPs caused a reduction in ROS generation, which is probably connected with the inhibition of apoptosis in these cells.

## 4. Material and Methods

### 4.1. Reagents

Dulbecco’s Modified Eagle Medium (DMEM) with GlutaMax, trypsin–EDTA, penicillin, and streptomycin, fetal bovine serum Gold (FBS Gold) were provided by Gibco (Paisley, UK), and Annexin V Apoptosis Detection Kit I by BD Pharmingen (San Diego, CA, USA). Dichlorodihydrofluorescein diacetate (DCFHDA), fumed silica powder 7 nm, silica dioxide nanopowder 5–15 nm, and silica dioxide nanopowder 10–20 nm were purchased from Sigma (St Louis, MO, USA).

### 4.2. Cell Culture Exposure to SiNPs

The LBC3 cell line was developed from glioblastoma multiforme tissue taken from a 56-year-old female patient subjected to surgical tumor resection [33], and was kindly given to us by Prof. Cezary Marcinkiewicz from the Department of Neuroscience, Temple University, Philadelphia, United States. The LN-18 cell line was provided by American Type Culture Collection (ATCC). Both cell lines were cultured in DMEM with GlutaMax, supplemented with heat-inactivated, 10% (FBS Gold), streptomycin (100 μg/mL), and penicillin (100 U/mL). The cells were cultured in Falcon flasks (BD) at 37 °C in a humidified atmosphere of 5% CO_2_, 95% air in an incubator Galaxy S+ (RS Biotech). At approximately 70% confluence, cells were detached with 0.05% trypsin and 0.02% EDTA in calcium-free, phosphate-buffered saline, and counted in a Scepter cell counter (Millipore). Then, 2.0 × 10^5^ cells were seeded in 2 mL of growth medium in six-well plates. In order to minimize the aggregation of SiNPs, prior to the experiments, nanoparticles were dispersed in deionized water by a sonicator (Sonopuls, Bandelin; 160 W, 20 kHz), on ice for 10 min. After 24 h, the medium was removed and replaced with the medium containing SiNPs suspensions at three different sizes (7 nm, 5–15 nm, or 10–20 nm), at concentrations ranging from 12.5 to 1000.0 µg/mL. The LBC3 and LN-18 cells not treated with SiNPs served as a negative controls. Next, the cells were incubated for 24 and 48 h and held for further analyses.

### 4.3. Characterization of SiNPs

Specific information regarding the characterization and physicochemical properties of the SiNPs in deionized water or medium have been described in our previous paper [34].

### 4.4. Cell Viability

Cell viability was evaluated according to the method of Carmichael et al. [35], using 3-(4,5-dimethylthiazol-2-yl)-2,5-diphenyltetrazolium bromide (MTT). Briefly, the LBC3 and LN-18 cells were seeded in six-well plates at a density of 2.5 × 10^5^ per well. After 24 h, the confluent cells were exposed to various concentrations of SiNPs ranging from 12.5 to 1000 µg/mL, at the three different sizes—7 nm, 5–15 nm, or 10–20 nm—and incubated for 24 h and 48 h. Next, the cells were washed three times with PBS and incubated with 1 mL of MTT solution (0.25 mg/mL in PBS), at 37 °C, in a humidified 5% CO_2_ atmosphere, for 4 h. Then, the medium was removed, and formazan products were solubilized in 1 mL of 0.1 mol/L HCl in absolute isopropanol. The absorbance of a converted dye in living cells was read on a microplate reader (Tecan) at λ = 570 nm. All experiments were run in duplicates in at least three cultures.

### 4.5. Detection of Apoptosis and Necrosis

Apoptotic and necrotic LN-18 and LBC3 cells were detected using an Fluorescein Isothiocyanate (FITC) Annexin V apoptosis detection Kit I, followed by flow cytometry on a FACSCanto II cytometer (Becton Dickinson) analysis. The cells were seeded in six-well plates at a density of 2.5 × 10^5^ per well, in 2 mL of growth medium, and incubated until they achieved confluency. After 24 h, the confluent cells were exposed to various concentrations of SiNPs, ranging from 25 to 600 µg/mL, at the three diverse sizes: 7 nm, 5–15 nm, or 10–20 nm, and incubated for 24 h and 48 h. Then, cells were detached by trypsinization and resuspended in DMEM, and subsequently in binding buffer. The cells were assayed using FITC Annexin V and propidium iodide (PI) staining, according to the manufacturer’s manual. Annexin V joins with high affinity to phosphatidylserine, and is used for identification of the cells in all stages of programmed cell death. Propidium iodide stains only disrupted cell membranes, and were used for identification of necrotic cells. FACSDiva software were used for data analysis. The dead cells were shut down on the basis of forward- and side-scatter parameters. The cells were recognized as early apoptotic (Annexin V+/PI−), or as a late apoptotic (Annexin V+/PI+). The sum of the Q2 and Q4 quadrants of scrutinized cells was submitted as a percentage of apoptotic cells.

### 4.6. ROS Detection

Detection of intracellular reactive oxygen species (ROS) was assessed using fluorescent dichlorodihydrofluorescein diacetate (DCFHDA). This compound enters the cell and is hydrolyzed, deacetylated by intracellular esterases to form a non-fluorescent compound, which is afterward oxidized by intracellular ROS forming the highly fluorescent compound 2′,7′–dichlorofluorescein (DCF). Briefly, the LBC3 and LN-18 cells were seeded in 96-well black plates at a density of 1 × 10^5^ cells per well, and were allowed to attach for 24 h. Then, the cells were treated with 10 μM of DCFHDA in PBS and incubated at 37 °C, in a 5% CO_2_ incubator, for 45 min. Next, the dye was removed, and cells were exposed to 50 and 100 µg/mL concentrations of SiNPs, at the three different sizes (7 nm, 5–15 nm, or 10–20 nm), and incubated for 24 h and 48 h. Subsequently, the cells were washed, trypsinized, and collected by centrifugation. The DCF fluorescence was quantified by an Infinite M200 microplate reader (Tecan, Salzburg, Austria), using the excitation wavelength λ = 485 nm and the emission wavelength λ = 535 nm. The intracellular ROS generation in silica nanoparticle-stimulated LBC3 and LN-18 cells was presented as the intensity of fluorescence of the DCF.

### 4.7. Statistical Analysis

STATISTICA version 13.3 program (StatSoft, Inc., Tulsa, OK, USA) and GraphPad Prism software version 9 (GraphPad Software, San Diego, CA, USA) were used for statistical analysis. Data were presented as means, standard deviations (SD), and percentages (%). Data were statistically analyzed by one-way analysis of variance (ANOVA). Duncan’s test was used as a post hoc test for the comparison of significance between groups. A *p* value < 0.05 was considered statistically significant.

## 5. Conclusions

Our results demonstrate that SiNPs show anticancer potential in glioblastoma cells. This effect is mainly achieved by the induction of two types of cell death: apoptosis, accompanied by augmented production of intracellular ROS, and necrosis, which is connected with reduction in ROS generation. These effects are highly cell type-specific and dependent on the size and dose of the nanoparticles. We have found size-dependent reduction in the viability of LBC3 cells, occurring in the following order: 5–15 > 10–20 > 7 nm. In contrast, the cytotoxicity of SiNPs in LN-18 cells was not affected by their size. This suggests that the same SiNPs can trigger different molecular pathways depending on cell type of glioblastoma and exposure conditions. In conclusion, our studies confirm that although a considerable amount research has already been conducted in the field of nanotoxicology, the reports are often contradictory, and more precise and accurate studies are still necessary to confirm the ways in which SiNPs might cause apoptotic or necrotic cell death in glioblastoma cells. While the use of silica nanoparticles in oncology opens up new perspectives for drug delivery and cancer therapy, there is a need to investigate the potential toxic effects and cellular damage caused by nanomaterials, which SiNPs cause in cancer cells.

## Figures and Tables

**Figure 1 ijms-22-03564-f001:**
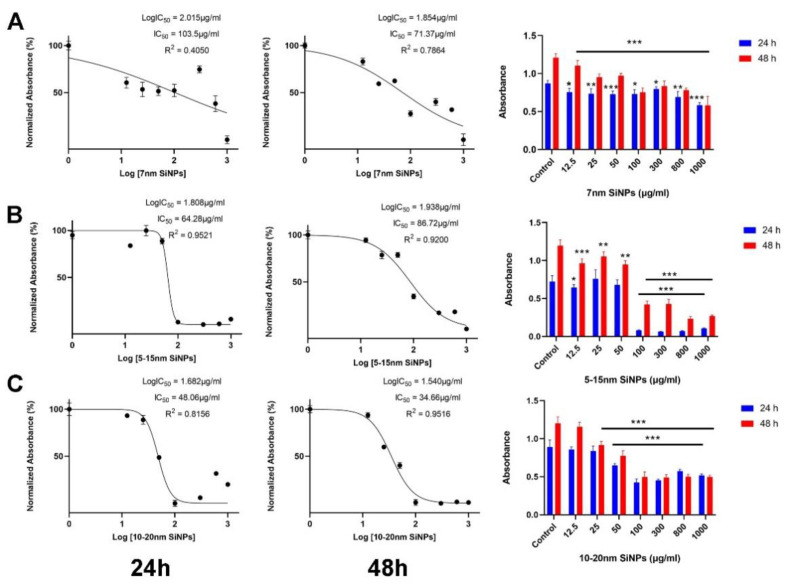
The viability of LBC3 cells (**A**–**C**) treated with different concentrations (12.5 to 1000 µg/mL) of silica nanoparticles in three different sizes—7 nm (**A**), 5–15 nm, (**B**) and 10–20 nm (**C**), for 24 and 48 h. Mean values from three independent experiments ± SD are presented. Note: Significant alterations are expressed relative to controls and marked with asterisks: * *p* < 0.05; ** *p* < 0.01; *** *p* < 0.001. Statistical significance was considered if *p* < 0.05.

**Figure 2 ijms-22-03564-f002:**
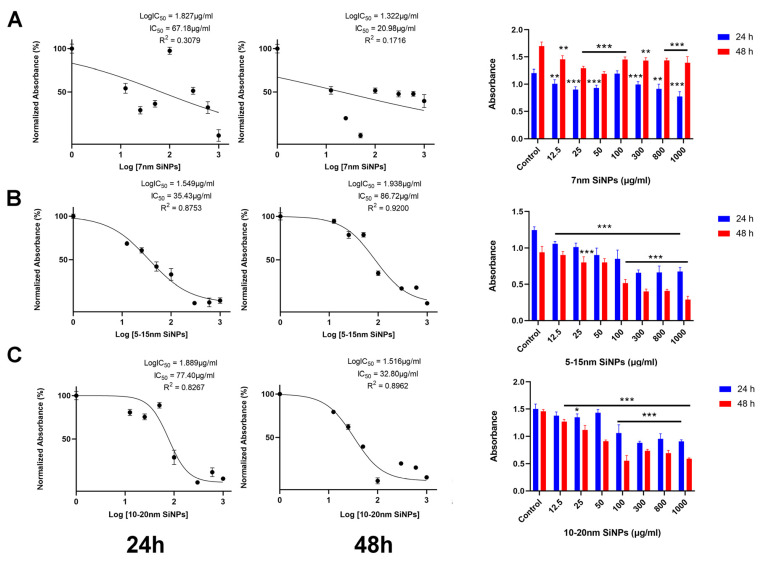
The viability of LN-18 cells (**A**–**C**) treated with different concentrations (12.5 to 1000 µg/mL) of silica nanoparticles in three different sizes—7 nm (**A**), 5–15 nm (**B**), and 10–20 nm (**C**)—for 24 and 48 h. Mean values from three independent experiments ± SD are presented. Note: Significant alterations are expressed relative to controls and marked with asterisks: * *p* < 0.05; ** *p* < 0.01; *** *p* < 0.001. Statistical significance was considered if *p* < 0.05.

**Figure 3 ijms-22-03564-f003:**
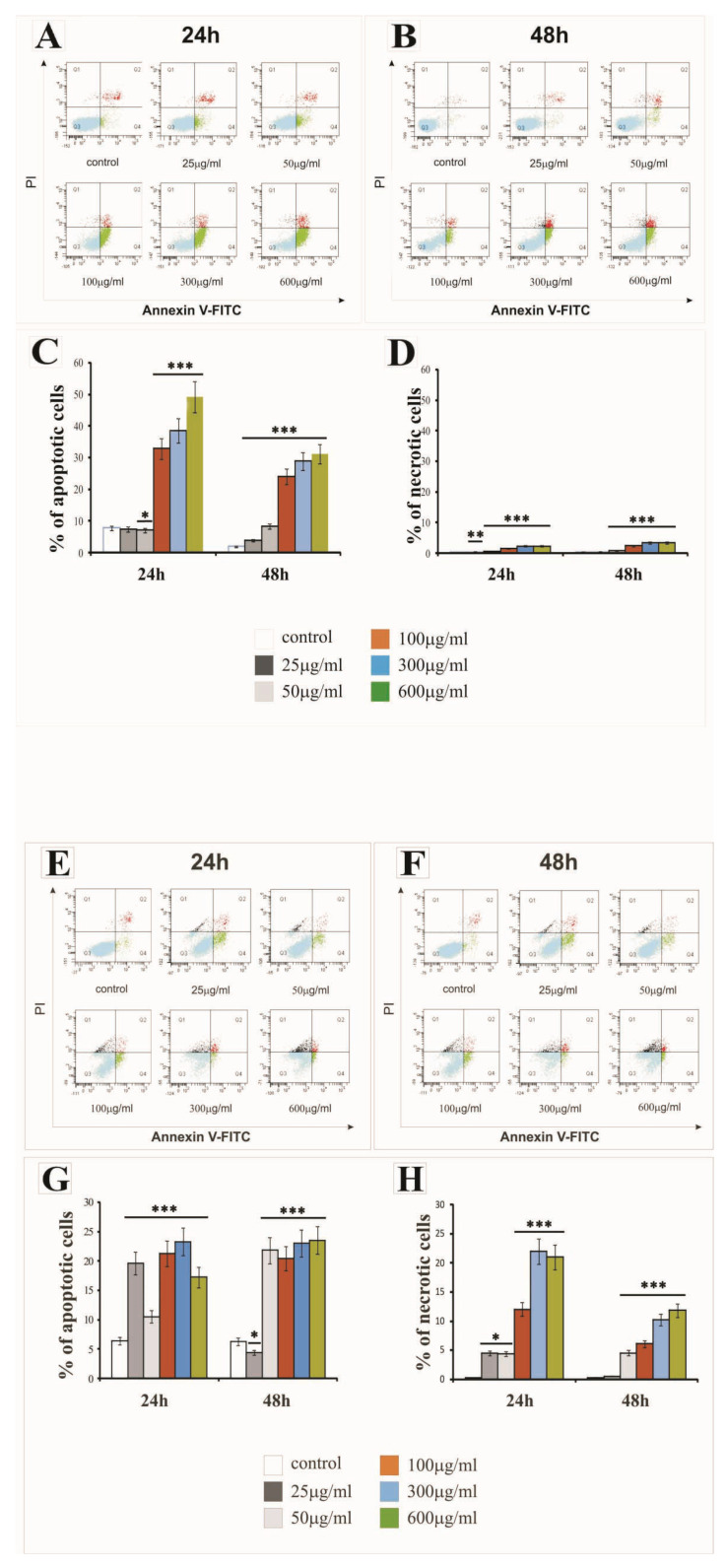
The effect of 7 nm silica nanoparticles on apoptosis and necrosis of LBC3 (**A**–**D**) and LN-18 (**E**–**H**) cells evaluated by annexin V assay. The cells were incubated for 24 and 48 h in Dulbecco’s Modified Eagle Medium (DMEM) with 25 μg/mL, 50 μg/mL, 100 μg/mL, 300 μg/mL, or 600 μg/mL of 7 nm silica nanoparticles. The cells were double-stained Fluorescein Isothiocyanate (FITC)-Annexin V and propidium iodide (PI). Representative Flow Cytometry (FACS) analysis via Annexin V-FITC/PI staining for 24 h and 48 h is presented. The bar graphs present the percentage of apoptotic cells as a sum of Q2 and Q4 quadrants (Figure 3C,G) and necrotic cells as a Q1 quadrant (Figure 3D,H) of the analyzed cell population. Mean values of the percentage of apoptotic and necrotic cells from three independent experiments ± SD are presented. Note: significant alterations are expressed relative to controls and marked with asterisks: * *p* < 0.05; ** *p* < 0.01; *** *p* < 0.001. Statistical significance was considered if *p* < 0.05.

**Figure 4 ijms-22-03564-f004:**
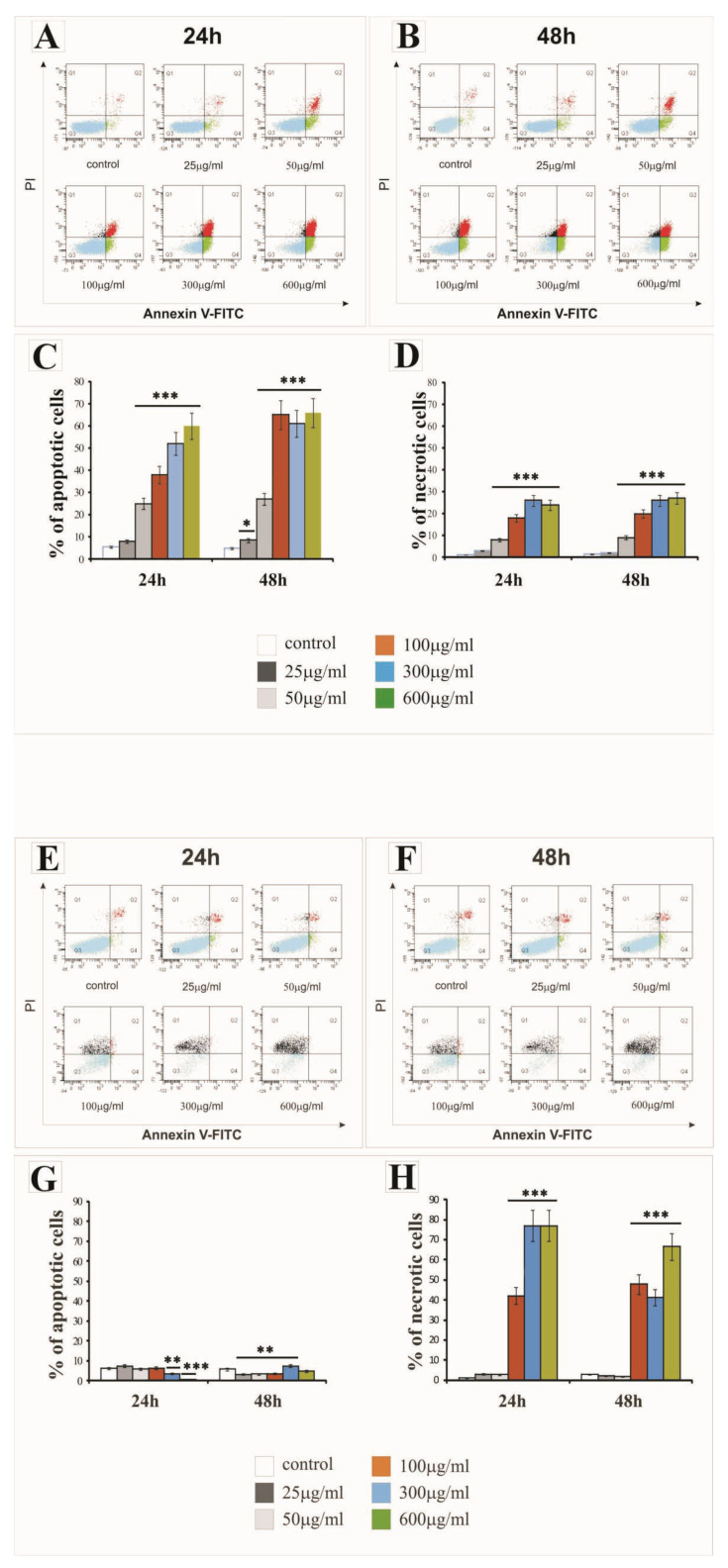
The effect of 5–15 nm silica nanoparticles on apoptosis and necrosis of LBC3 (**A**–**D**) and LN-18 (**E**–**H**) cells evaluated by Annexin V assay. The cells were incubated for 24 and 48 h in DMEM with 25 μg/mL, 50 μg/mL, 100 μg/mL, 300 μg/mL, or 600 μg/mL of 5–15 nm silica nanoparticles. The cells were double-stained with FITC–Annexin V and PI. Representative FACS analysis via Annexin V-FITC/PI staining for 24 h and 48 h is presented. Bar graph presenting the percentage of apoptotic cells as a sum of Q2 and Q4 quadrants (Figure 4C,G) and necrotic cells as a Q1 quadrant (Figure 4D,H) of analyzed cells population. Mean values of the percentage of apoptotic and necrotic cells, from three independent experiments ± SD are presented. Note: significant alterations are expressed relative to controls and marked with asterisks: * *p* < 0.05; ** *p* < 0.01; *** *p* < 0.001. Statistical significance was considered if *p* < 0.05.

**Figure 5 ijms-22-03564-f005:**
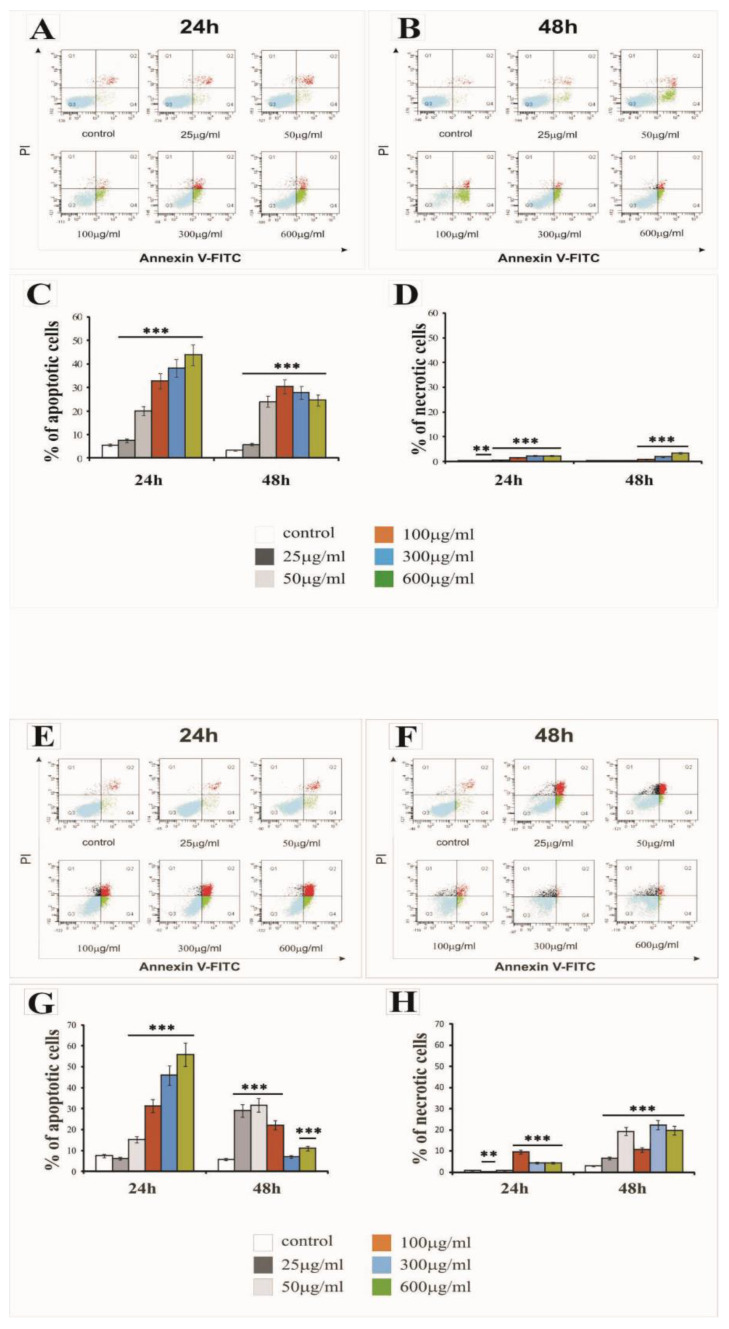
The effect of 10–20 nm silica nanoparticles on apoptosis and necrosis of LBC3 (**A**–**D**) and LN-18 (**E**–**H**) cells evaluated by Annexin V assay. The cells were incubated for 24 and 48 h in DMEM with 25 μg/mL, 50 μg/mL, 100 μg/mL, 300 μg/mL, or 600 μg/mL of 10–20 nm silica nanoparticles. The cells were double-stained with FITC-Annexin V and PI. Representative FACS analysis via Annexin V-FITC/PI staining for 24 h and 48 h is presented. Bar graphs present the percentage of apoptotic cells as a sum of Q2 and Q4 quadrants (**C**,**G**), necrotic cells as a Q1 quadrant (**D**,**H**) of the analyzed cell population. Mean values of the percentage of apoptotic and necrotic cells from three independent experiments ± SD are presented. Note: significant alterations are expressed relative to controls and marked with asterisks: ** *p* < 0.01; *** *p* < 0.001. Statistical significance was considered if *p* < 0.05.

**Figure 6 ijms-22-03564-f006:**
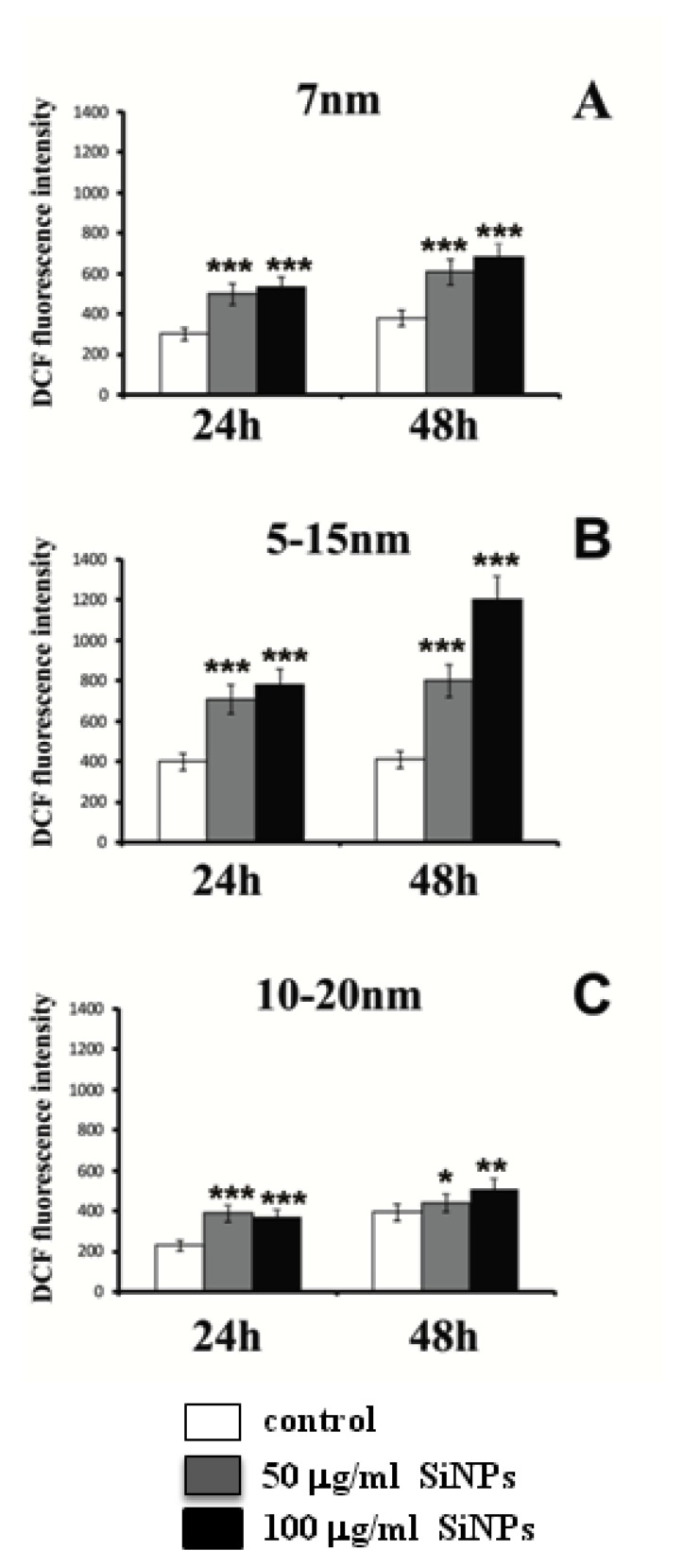
Intracellular reactive oxygen species (ROS) production induced by three different sizes—7 nm (**A**), 5–15 nm (**B**), and 10–20 nm (**C**)—of silica nanoparticles (SiNPs) in LBC3 cells for 24 and 48 h. Note: significant alterations are expressed relative to controls and marked with asterisks: * *p* < 0.05; ** *p* < 0.01; *** *p* < 0.001. Statistical significance was considered if *p* < 0.05.

**Figure 7 ijms-22-03564-f007:**
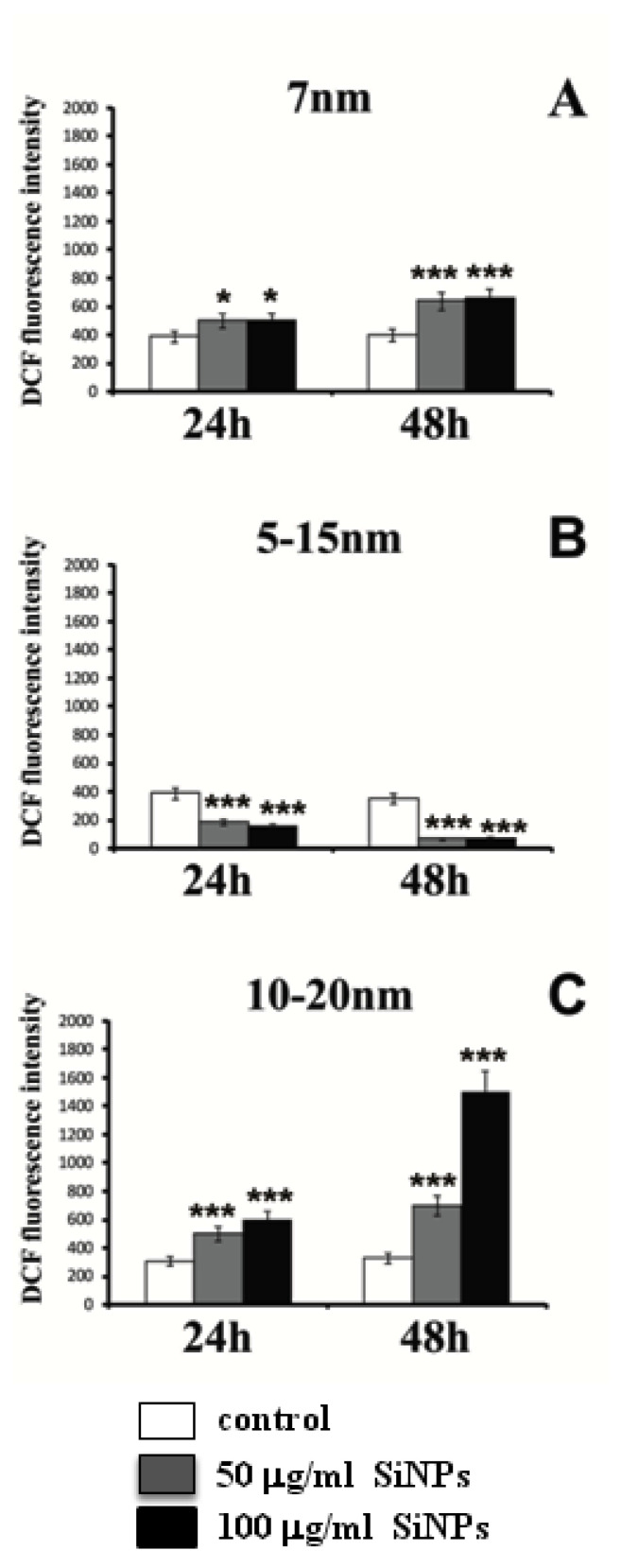
Intracellular reactive oxygen species (ROS) production induced by three different sizes—7 nm (**A**), 5–15 nm (**B**) and 10–20 nm (**C**)—of SiNPs in LN-18 cells for 24 and 48 h. Note: significant alterations are expressed relative to controls and marked with asterisks: * *p* < 0.05; *** *p* < 0.001. Statistical significance was considered if *p* < 0.05.

## Data Availability

Not applicable.

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
