# Peer review of "The Pro-Apoptotic Effect of Silica Nanoparticles Depends on Their Size and Dose, as Well as the Type of Glioblastoma Cells"

_ijms, 2021, doi:10.3390/ijms22073564_

Round 1

Reviewer 1 Report

.

Author Response

Thank you kindly again for your comments and suggestions. Thanks for your being interested in our work. Your valuable advices were very important to improve the quality of this manuscript.

Reviewer 2 Report

The resubmitted manuscript was significantly improved and all the comments were addressed correctly. The article can be accepted at the present form. 

Author Response

(The authors gave the same response as above.)

Reviewer 3 Report

The authors have addressed my main concern about data duplication between the current manuscript and the previously published report in Nanomaterials 2017. Although the current work has indeed provided a more detailed information on the impact of size and concentration of SiNPs on different glioblastoma cell lines, the underlying molecular mechanism triggering cell death should be documented in more details. The previous work in Nanomaterials 2017 provides a solid evidence on mitochondrial-dependent apoptosis in LBC3 cells, something that is missing in the current manuscript. The paper would further improve if more mechanistic studies are included (in addition to ROS), e.g. WB analysis of apoptosis vs necrosis (e.g PARP, caspase,  survivin, Bcl2, BIM...) in LN-18 cells treated with 5-15nm SiNPs to provide a further evidence that apoptosis is not activated in those conditions. I also have a concern regarding the flow cytometry data analyses. The authors determined the number of apoptotic cells as a sum between Annexin V+ and Annexin V+/PI+ cells (Q2 + Q4), which is not appropriate. The Annexin V+/PI + cells correspond to late apoptotic/necrotic cells and should instead be counted as one. To consider the paper for publication, please re-quantify the data accordingly and discuss the obtained results. 

Round 2

Reviewer 3 Report

I believe that the mechanistic studies would greatly improve this manuscript and hope to see the results in the future. Although I do not agree on the issue of apoptosis determination, I would leave this point since as pointed out by the authors the flow data might be interpreted in different ways by different groups. To me the true apoptotic population corresponds to annexin V+ only cells, while the double stained population in addition to late apoptotic cells also includes the necrotic cells (as demonstrated in many other studies, e.g. Gill et al. Mol Cancer 2009, doi: 10.1186/1476-4598-8-39). 

This manuscript is a resubmission of an earlier submission. The following is a list of the peer review reports and author responses from that submission.

Round 1

Reviewer 1 Report

In this study Krҿtowski et al analyse the in vitro effect of silica nanoparticules (SiNPs) on the viability of two different gliobastoma cell lines observing an apoptotic and/or necrotic response depending on the  size and dose of SiNPs  proposing a role for ROS generation.

However different questions arise from this work

  • The molecular and cellular mechanisms underlying the effects of SiNPs on glioblastoma cells were not resolved
  • The effect on ROS production should be analysed at short term cultures
  • Is the apoptotic/ necrotic effect of SiNPs restricted to cancer or proliferating cells?
  • Why do LBC3 and LN-18 cells response differently?
  • Do glioblastoma cells endocyte SiNPs?
  • Which is the glioblastoma cell: SiNP ratio in each experiment?

Reviewer 2 Report

The article is dedicated to the study of the silica nanoparticles influence on the death of glioblastoma cells and provides descriptive insights into SINPs of different size actions on glioblastoma cells.

The article is well written but due to the large data amount is quite hard to follow. Mechanistic insights provided within the study require concretization.

Some points should be considered before article acceptance

  1. Data on Figs 1 and 2 seem to be previously published (https://doi.org/10.3390/nano7080230). It should be at least reanalyzed to provide new information about different SINPs activity in GBM cells:
  2. Non-linear regression should be done and SINPs IC50 should be determined by Graphpad prism or any online tool. Regression curves of different SINPs action on LCB3 and LN-18 cells should be compared by f-test. IC50 and A1 (curve fit bottom) parameters should be presented as a supplementary table.
  3. As data was normalized to untreated cells viability, the t-test should be replaced by One sample t-test. Also, for all figures, the significance should be differentiated (* for p<0.05, ** for p<0.01, and so on).
  4. What are the duplication times of the cells?
  5. Why 10-20 nm SINPs action on LCB3 cells during 24 and 48 h incubation differs significantly, while the antiproliferative activity of that type of SINPs in LN-18 cells are principally the same?
  6. The hypothesis explaining the difference of SINPs action on LCB3 and LN-18 cells should be formed – what is the p53 status of cells? Maybe there is some difference in kinases or receptor expression in those cells? The difference in ROS generation by cells with apoptotic and necrotic cell death points to mitochondrial events as on critical in death type choose but what molecules can determine the type of cell death?
  7. What is the target of SINPs in glioma cells? Do the SINPs incorporate in cells by endocytosis? What type of apoptosis is induced in glioma cells? If the ROS generation has increased the mitochondria seem to participate in apoptosis, so you should determine the mitochondrial potential and cytochrome с activation in cells by WB or ICH.
  8. The biological effect of SINPs on glioma cells should be compared with that on primary/non-transformed astrocytes (mouse, rat, or human fetal).
  9. Bioavailability of SINPs, their clearance, and possible passage through BBB should be discussed to justify the prospects of SINPs clinical usage.
  10. For figures 3-6 One-way ANOVA followed by Dunnets test should be used to compare the effects of different SNIPs concentration with untreated cells.
  11. For figures 6 and 7 provide legend with SINPs concentration.
  12. The figures are too big. Please rearrange the panels to make figures compact and place all figure legends under the pictures.

Altogether the article is interesting and gives us new insights into SINPs biological effects and can be considered for publication after revision, strengthening its mechanistic observations and clinical perspective.

Reviewer 3 Report

In this work authors aim at investigating the molecular mechanisms of glioblastoma cell death induced by silica nanoparticles. Although the data seem sound, the vast majority of the findings have already been published by the group in Nanomaterials (Basel) 2017 Aug; 7(8): 230. (doi: 10.3390/nano7080230) and therefore do not bring any novelty do the field.

The aforementioned published work (Nanomaterials 2017) extensively describes the mechanisms of cell-death induction by silica nanoparticles, reaching beyond the current manuscript.

As such, I'm compelled to reject this work based on novelty, scientific significance and data duplication/self-plagiarism (in case of Figure 1&2).